# A Novel Sparsity Adaptive Algorithm for Underwater Acoustic Signal Reconstruction

**DOI:** 10.3390/s22135018

**Published:** 2022-07-03

**Authors:** Na Li, Xinghui Yin, Haitao Li

**Affiliations:** 1College of Mechanical and Electronic Engineering, Suzhou University, Suzhou 234000, China; lina@ahszu.edu.cn; 2College of Computer and Information, Hohai University, Nanjing 211100, China; xhyin@hhu.edu.cn

**Keywords:** underwater acoustic signal reconstruction, sparsity adaptive, variable step size, least squares method, sonar system

## Abstract

In view of the fact that most of the traditional algorithms for reconstructing underwater acoustic signals from low-dimensional compressed data are based on known sparsity, a sparsity adaptive and variable step-size matching pursuit (SAVSMP) algorithm is proposed. Firstly, the algorithm uses Restricted Isometry Property (RIP) criterion to estimate the initial value of sparsity, and then employs curve fitting method to adjust the initial value of sparsity to avoid underestimation or overestimation, before finally realizing the close approach of the sparsity level with the adaptive step size. The algorithm selects the atoms by matching test, and uses the Least Squares Method to filter out the unsuitable atoms, so as to realize the precise reconstruction of underwater acoustic signal received by the sonar system. The experimental comparison reveals that the proposed algorithm overcomes the drawbacks of existing algorithms, in terms of high computation time and low reconstruction quality.

## 1. Introduction

The traditional underwater acoustic array takes the cross-correlation output of the signal received by the sonar system as the test statistic of the reconstruction algorithm [1]. However, due to the influence of multi-path propagation and noise in the real wave-guide environment, the attenuation phenomenon of the spatial correlation of the received signals of different array elements is extremely easy to produce [2,3]. How to improve the correlation of the received signal to achieve precise reconstruction of the underwater array is a research hot spot. Li proposed a frequency shift compensation method based on wave-guide invariant to improve the correlation of signals received by the sonar system [4,5]. However, the wave-guide invariant will change with the topography in complex terrain. This problem can be improved by the Compressed Sensing (CS) theory, which can reduce the distortion of the signal effectively by setting a reasonable threshold to remove the small characteristic part and retain the essential characteristic part of the signal.

CS theory, proved by Emmanuel Candes et al., pointed out that a sparse or compressible signal could be reconstructed exactly in some transform domains [6]. CS method is an emerging signal processing theory on sampling, which allows the acquiring and reconstruction of signals using sub-Nyquist measurements [7,8]. Recovering the original signal from the compressed observation data is a convex optimization problem, and solving a linear optimization problem based on the norm is an NP-hard problem. It can be solved by using a greedy algorithm with progressive approximation strategy, in which the orthogonal matching pursuit (OMP) algorithm is the most representative [9]. Subsequently, scholars have studied many improved algorithms based on orthogonal matching pursuit algorithm [10,11,12]. Though the algorithms mentioned above can improve the spatial correlation of underwater acoustic signal and realize the detection and estimation of underwater acoustic signal, they need sparsity as a prior knowledge. Thong T.DO developed a sparse adaptive matching pursuit (SAMP) algorithm in 2009 [13], but the fixed step size of SAMP is not flexible enough to avoid overestimation and underestimation. Most of the improved algorithms employ the method of reducing step-size if the algorithm starts to converge, however the improvement effect is not ideal. A sparsity adaptive subspace tracking (SASP) algorithm was proposed [14], which has many iterations and high computational complexity. Therefore, the algorithm is not feasible in the underwater target detection system with high real-time requirement.

Therefore, the algorithm of underwater acoustic signal reconstruction based on sparsity adaptive needs further research and exploration, it is very important for the signal processing to find an algorithm that takes into account both the reconstruction time and reconstruction accuracy.

## 2. Theoretical Analysis

### 2.1. Compressed Sensing

Assume that a real valued and discrete signal x with *P* sparsity can be represented by *P* significant coefficients over an N-dimensional basis, where $P\leq N,x\in R^{N}$[15]. Ignoring background noise, the compressed sensing could be described as recovering a high-dimensional signal x from a low-dimensional signal y:(1)$$y_{M\times 1}=\Phi_{M\times N}x_{N\times 1}$$
where $y\in R^{M}$, $\Phi_{M\times N}$ is the sensing matrix with more columns *N* than rows *M*. Generally, reconstructing x from y can be regarded as a problem of solving the equation whose solution is infinite. Considering the sparse constraint of x, the reconstruction problem can be described as the following minimization [16]:(2)$$\arg\min_{x}{∥x∥}_{0}\begin{array}{cc} & s.t.\begin{array}{cc} & y=\Phi x \end{array} \end{array}$$
where ${∥x∥}_{0}$ is the $l_{0}$ norm of **x**. Since the instability of ${∥x∥}_{0}$, the solution of the Equation (2) is not convergent. The $l_{0}$-norm minimization problem is NP-hard, which is solved by selecting atoms to the support set of the unknown signal sequentially in the greedy algorithm. Another solution method is to convert the Equation (2) to an $l_{1}$-norm minimization one:(3)$$\arg\min_{x}{∥x∥}_{1}\begin{array}{cc} & s.t.\begin{array}{cc} & y=\Phi x \end{array} \end{array}$$

This kind of reconstruction is called as convex optimization algorithm, which is performs better in the stability. However, the algorithm run-time becomes longer as the dimension increases. Many improved greedy methods such as SAMP and SASP algorithms perform better in both stability and run-time. Moreover, the RIP criterion has been proposed for Φx to guarantee that Φx has the unique solution corresponding to the measurement vector. The sensing matrix Φx must satisfy the RIP criterion of order *P* if there $\delta_{P}\in \left(0,1\right)$) such that
(4)$$\left(1-\delta_{P}\right){∥x∥}_{2}^{2}\leq {∥\Phi x∥}_{2}^{2}\leq \left(1+\delta_{P}\right){∥x∥}_{2}^{2}$$
for any *P*-sparse vector x(${∥x∥}_{0}^{}\leq P$).

Greedy algorithm and convex optimization algorithm are the two main methods of signal reconstruction as mentioned above. Unlike the convex optimization method, the greedy method does not constrain the sparsity of the signal, but directly seeks the sparsity solution by means of matching. This algorithm has many advantages, such as simple implementation, low computational complexity and fast convergence speed. It meets the requirement of underwater acoustic signal processing for the sonar system.

### 2.2. Ship Radiated Noise

In this paper, the corresponding mathematical model of the underwater acoustic signal received by the sonar system is established according to the characteristics of the frequency spectrum. The propeller and its own motor and other propulsion devices, as well as the ship body shuttle water movement, will produce noise radiation in the water when the ship is sailing. According to the noise source, ship-radiated noise has three types, propeller noise, hydrodynamic noise and mechanical noise, whose frequency spectrum includes a line spectrum and continuous spectrum.

The line spectrum is mainly formed by mechanical vibration, propeller resonance and blade cutting flow. The frequency $f_{m}$ of the line spectrum is related to the ship speed *v* and the propeller blades number *l*, which satisfies the following relationships [17]
(5)$$f_{m}=mvl$$
where *m* denotes the number of harmonics.

The continuous spectrum is mainly caused by the wide-band noise generated by the cavitation of the propeller, which is caused by the random breakdown of a large number of bubbles near the propeller. The mathematical model of ship-radiated noise x(t) can be expressed as
(6)$$x\left(t\right)=x_{c}\left(t\right)+x_{l}\left(t\right)$$
where $x_{c}\left(t\right)$ and $x_{l}\left(t\right)$ are the time domain expression of continuous spectrum and line spectrum.

There are many algorithms for signal reconstruction. The two main methods are the greedy algorithm and convex optimization algorithm, as mentioned above. Unlike the convex optimization algorithm, the greedy algorithm does not restrict the sparse nature of the signal. Rather, it uses the matching method to find it.

## 3. SAVSMP Algorithm

In order to overcome the shortcomings of the SAMP and SASP algorithms in underwater acoustic signal reconstruction, a sparse adaptive variable step size matching pursuit (SAVSMP) algorithm is proposed for the sonar system.The flow chart of the main procedure is shown in Figure 1. The algorithm includes initial sparsity estimation, first selecting, candidate set making, second selecting, support set updating, residual updating and step-size setting parts. The novelty of the SAVSMP algorithm is that new methods are employed in initial sparsity estimation part, first selecting part and step-size setting part.

### 3.1. Initial Sparsity Estimation

The flow chart of the initial sparsity estimation is shown in Figure 2.

There are several stages in the recovery process, and each stage includes several iterations. Here, *i* denotes the iteration index, and *j* indicates the stage index. Moreover, $r_{0}$ and $K_{0}^{{}^{\prime }}$ are denoted as the initial residual and initial sparsity. The initial sparsity is similar to SASP and AStMP algorithms, which select atoms by the RIP criterion. The criterion is formally defined as Theorem 1, which is proved by [14].

**Theorem** **1.**
*If the sensing matrix*
**Φ**
*satisfies the RIP criterion with the parameters $\left(P,\delta_{P}\right)$, when $K_{0}^{{}^{\prime }}\geq P$, one has*

(7)
$${∥\Phi_{F_{K_{0}^{{}^{\prime }}}}*y∥}_{2}\geq \frac{\left(1-\delta_{P}\right)}{\sqrt{\left(1+\delta_{P}\right)}}{∥y∥}_{2}$$

*where $F_{{}_{K_{{}_{0}}^{{}^{\prime }}}}^{}$ is the support set consisting of ${K_{0}}^{{}^{\prime }}$ selected atoms. Furthermore, we have Theorem 2 as follows.*


**Theorem** **2.**
*If the sensing matrix*
**Φ**
*satisfies the deformation of the RIP criterion with the parameters $\left(P,\delta_{P}\right)$ when $K_{0}^{{}^{\prime }}≺P$, one has*

(8)
$${∥\Phi_{F_{{}_{K_{0}^{{}^{\prime }}}}}*y∥}_{2}\leq \frac{\left(1-\delta_{P}\right)}{\sqrt{\left(1+\delta_{P}\right)}}{∥y∥}_{2}$$



If Equation (8) is satisfied, the initial sparsity $K_{0}^{{}^{\prime }}$ = $K_{0}^{{}^{\prime }}$ +1. The support set will be updated as
(9)$$F_{K_{0}^{{}^{\prime }}}=\max(|\Phi *y|,K_{0}^{{}^{\prime }})$$

After updating, and algorithm continuous to return to the judgment part. It indicates that the initial sparsity estimation is completed when the criterion is not met.

It can be seen from the Equation (8) that the larger the $\delta_{P}$ is, the smaller the $K_{0}^{{}^{\prime }}$ would be; on the contrary, the smaller the $\delta_{P}$ is, the larger the $K_{0}^{{}^{\prime }}$ that could be obtained. Therefore, $K_{0}^{{}^{\prime }}$ should be adjusted to $K_{0}$ in order to prevent over-estimation and under-estimation of the sparsity estimation. Taking $\delta_{P}$ and $K_{0}/K_{{}_{0}}^{{}^{\prime }}$ as the independent variable and dependent variable, respectively, the function can be obtained by the MATLAB curve fitting, $K_{0}=(a_{1}*\delta_{P}$ + $b_{1})*K_{{}_{0}}^{{}^{\prime }}$, $a_{1}=10,b_{1}=-0.5$. The Sum of Squares due to Error (SSE) takes the value of 4.437 × 10${}^{-31}$, which means the goodness-of-fit of the obtained formula is high.

The over-estimated value is reduced when $\delta_{P}$ is smaller, and the under-estimated value is increased when $\delta_{P}$ is larger. However, $K_{0}^{{}^{\prime }}$ is halved whatever value the RIP constant $\delta_{P}$ takes in the AStMP algorithm which will result in under-estimation. After updating the support set and the residual, the subprogramme returns.

### 3.2. First Selecting

The greedy algorithms usually use the inner product criterion to measure the similarity between vectors. Suppose $x=(x_{1},x_{2},\ldots ,x_{n})$, and $y=(y_{1},y_{2},\ldots ,y_{n})$ are two n-dimensional vectors, the inner product criterion between them can be defined as follows [18]:(10)$$\cos\left(x,y\right)=\frac{\langle x,y\rangle }{∥x∥\cdot ∥y∥}=\frac{\sum_{t=1}^{n}x_{t}y_{t}}{{\left(\sum_{t=1}^{n}x_{t}^{2}\sum_{t=1}^{n}y_{t}^{2}\right)}^{1/2}}$$

It can be seen that the inner product criterion realizes the selection of atoms by calculating the angle cosine of the sensing matrix and the residual. The greater the absolute value, the more relevant the residual is to the selected atom from the sensing matrix. However, this matching method can not reflect the effect of the important components of the amplified data when measuring the similarity of signals [19]. Therefore, a better method needs to be found to select the atom that best matches the residual.

The Dice coefficient criterion is introduced here, and its definition is as follows [20,21,22]:(11)$$\mathrm{Dice}\left(x,y\right)=\frac{\sum_{t=1}^{n}x_{t}y_{t}}{(\sum_{t=1}^{n}x_{t}^{2}+\sum_{t=1}^{n}y_{t}^{2})/2}$$

It can be seen from Equations (10) and (11), cos(x,y) and Dice(x,y) have the same range of values [0, 1], and will be 1 when the two vectors are equal. It is easy to find that the denominator of the inner product criterion is ${\left(\sum_{t=1}^{n}x_{t}^{2}\sum_{t=1}^{n}y_{t}^{2}\right)}^{1/2}$ which represents the geometric average of the square sum calculated by the two vectors, while the denominator of the Dice coefficient criterion is $\left(\sum_{t=1}^{n}x_{t}^{2}+\sum_{t=1}^{n}y_{t}^{2}\right)/2$ which represents the arithmetic average of the square sum calculated by the two vectors. According to the average theory, in general, the geometric mean focuses on the average change trend of the overall sample, while the arithmetic average represents the unbiased estimate of the individual expectations. Consequently, the arithmetic mean is better to avoid partial information loss and preserve the integrity of the signal [23]. Therefore, the Dice coefficient criterion is more conducive to select the appropriate atom than the inner product criterion, it can reflect the important role of each element of the signal [24,25]. Therefore, the proposed algorithm SAVSMP uses, the following formula to match the best atoms [26,27]:(12)$$\Omega_{i}=\max\left(\mathrm{Dice}\left(\Phi ,r_{i-1}\right),K_{j}\right)$$

### 3.3. Step-Size Setting

Adaptive step size is used in the The SAVSMP algorithm, which will reduce the running time effectively. The residual r is compared with the threshold $\sigma_{1}$, firstly. If  ${∥r∥}_{0}\leq \sigma_{1}$, the operation of the program would be terminated, Otherwise, the residuals between the two adjacent iterations would be compared. If  ${∥r∥}_{0}\geq {∥r_{i-1}∥}_{0}$, calculate the value of χ, which is defined as $\chi ={∥y∥}_{2}/{∥r∥}_{2}$. At the initial iteration of the greedy algorithms, the value of χ is equal to 1 since the residual r is assigned as y at first. The residual r decreases gradually as the number of iterations increases, and the iteration will end when r reaches the threshold. Therefore, it can be concluded as χ∈[1,+∞). This is because most of the effective atoms are put into the support set when χ is about 10 [28,29], ξ=9+1/χ, where ξ∈(9,10] is set as the threshold to choose the method for the step-size updating. SAVSMP uses the exponential curve fitting method if χ is smaller than ξ; otherwise, it employs the weak matching method to update the step-size. The flow chart of the step-size setting is shown in Figure 3.

In the exponential curve fitting method, the residual energy ratio χ and the step-size scaling ratio β are set as the independent variable and dependent variable. According to the expected change relation that β decreases exponentially with the increase of χ, the function $k_{j}=\left\lfloor \beta *k_{j-1}\right\rfloor$, $\beta =a_{2}*\exp\left(b_{2}*x\right)$, where $a_{2}\in \left(4.80,5.42\right),b_{2}\in \left(-0.26,-0.22\right)$ is obtained by Matlab simulation.The SSE of the function is about 0.011, which means the goodness-of-fit of the obtained function is high. The relation between χ and χ is shown in Figure 4, it can be seen that the step size changes in a nonlinear manner instead of undergoing linear growth.

A weak mating method is employed to realize the close approach of the sparsity level. The algorithm selects the maximum value $M_{a}$ and the intermediate value $M_{e}$ of the inner product v between the observation matrix $M_{a}$ and the residual $M_{a}$ to determine the threshold setting. The definitions of **v**, $M_{a}$ and $M_{e}$ are shown below:(13)$$V=\Phi *r_{i}V=\left(v_{1},v_{2},\cdots v_{n},\cdots \right)$$
(14)$$M_{a}=\max_{1<n<N}\left|v_{n}\right|$$
(15)$$M_{e}=\underset{1<n<N}{\mathrm{median}}\left|V_{n}\right|$$

The threshold γ is set as $gamma=\tau \left(a_{3}M_{a}+b_{3}M_{e}\right)/c_{3}$, where τ∈(0,1], $a_{3}$, $b_{3}$, and $c_{3}$ are set as 2, 3, and 3, respectively, considering a suitable threshold γ could help to select atoms more efficiently. Atoms are selected and put into the supporting set when $\left|v_{n}\right|>\gamma$. The step-size can be set by the parameter τ, and τ∈[0.8,0.9] can give consideration to both reconstruction accuracy and computation speed [28]. and the selected atoms are put into the supporting set. The step size of each stage can be adjusted by τ, and the experimental results show that the performance of the reconstruction and the computation speed can be considered at the same time when τ∈[0.8,0.9].

## 4. Performance Analysis of SAVSMP Algorithm

SAVSMP algorithm includes two aspects of initial sparsity estimation and signal reconstruction. The use of Dice coefficient criterion in SAVSMP algorithm facilitates the selection of better-matched atoms. In addition, with the idea of step size changing adaptively, the reconstruction accuracy of SAVSMP is comparable to that of SAMP with a fixed step size. The using of the initial sparsity estimation in SAVSMP is equivalent to reducing the sparsity of the target signal indirectly. The complexity of the initial sparsity estimation is mostly concentrated in the calculation of M times projections, so it is relatively small. However, reconstructing the target signal through the least squares method accounts for a significant portion of the computation. As a result, the whole complexity of the SAVSMP is determined by the stage number. Each stage can be considered as a single SP algorithm with a small sparsity. The proposed SAVSMP algorithm is summarized in Algorithm 1.

The proposed SAVSMP algorithm is summarized in Algorithm 1. It can be seen that the computation complexity of initial sparsity estimation part is mainly focused on the projection operation of Equation (9). The computation complexity of signal reconstruction part is mainly concentrated in the projection operation of Equation (12), while the signal reconstruction using least squares only accounts for a small amount of computation. Therefore, the computational complexity of SAVSMP algorithm depends on the computation of Equations (9) and (12). According to the analysis, Equation (9) needs $K_{0}^{{}^{\prime }}-\left\lceil lnM\right\rceil$ iterations and Equation (12) needs $\frac{P-K_{{}^{{}_{0}}}}{k_{SAVSMP}}$ iterations. Because the computation complexity of one projection operation is O(MN), the computation complexity of SAVSMP algorithm is $O(K^{{}_{0}^{{}^{\prime }}}-\left\lceil lnM\right\rceil +\frac{P-K_{0}}{k_{SAVSMP}})MN$. Since the step size increases at a larger slope before the threshold and linearly at a smaller slope after the threshold, $K^{{}_{0}^{{}^{\prime }}}-\left\lceil lnM\right\rceil +\frac{P-K_{0}}{k_{SAVSMP}}⩽\frac{P}{k_{{}_{SAMP}}}$ is obtained. Comparing the computational complexity of the OMP Algorithm O(PMN) [9], the SAMP algorithm $O(\frac{P}{k_{SAMP}}MN)$[13] and the SASP algorithm $O(P^{{}^{2}}MN)$ [14], it can be concluded that the proposed SAVSMP algorithm has lower computational complexity.

Each stage of the proposed algorithm can be considered as the SP algorithm which can reconstruct the signal via finite iterations [10]. Therefore, SAVSMP can also reconstruct x from y via finite stages if Φ satisfies the RIP criterion with the parameter $\delta_{P}<0.165$. Since ${∥r∥}_{{}_{2}}$ is monotonically decreasing, the algorithm will converge to a local minimum.


**Algorithm 1** Process of the SAVSMP Algorithm.**Input:** Measurement vector **y**, Sensing matrix Φ, RIP constant $\delta_{P}$,      parameter $\sigma_{1}$,$\sigma_{2}$, τ.
**Initialization:**

$r_{0}=y,K_{{}_{0}}^{{}^{\prime }}=\left\lfloor \ln M\right\rfloor ,i=0,j=0$

      **If** ${∥\Phi_{F_{{}_{K_{0}^{{}^{\prime }}}}}*y∥}_{2}\leq \frac{\left(1-\delta_{p}\right)}{\sqrt{\left(1+\delta_{p}\right)}}{∥y∥}_{2}$            $K_{{}_{0}}^{{}^{\prime }}=K_{{}_{0}}^{{}^{\prime }}+1$,$F_{K_{0}^{{}^{\prime }}}=\max(|\Phi *y|,K_{0}^{{}^{\prime }})$      **end if**            $K_{0}=(a_{1}*\delta_{P}+b_{1})*K_{{}_{0}}^{{}^{\prime }}$, $F_{0}=\max(|\Phi_{F_{K_{0}}}^{}*y|,K_{0})$,            $r_{0}=\min{∥y-{\Phi_{F}}_{{}_{0}}{x_{F}}_{{}_{0}}∥}_{2}$**repeat:**i=i+1,$\Omega_{i}=\underset{K_{j}}{\arg\max}\left|D(\Phi ,r_{i-1})\right|$,$H_{i}=F_{i-1}\cup \Omega_{i}$,            $F_{i}=\underset{K_{j}}{\arg\max}\left|\Phi_{H_{i}}^{+}y\right|$, $r_{i}=y-\Phi_{F_{i}}\Phi_{F_{i}}^{+}y$      **if** ${∥r_{i}∥}_{2}\leq \sigma_{1}$            quit the iteration.      **else if** ${∥r_{i}∥}_{2}\geq {∥r_{i-1}∥}_{2}$            $\chi ={∥y∥}_{2}/{∥r∥}_{2}$,ξ=9+1/χ            **if** $j\leq \sigma_{2}$                  **if**χ≺ξ                        $\beta =a_{2}*\exp\left(b_{2}*x\right)$, $k_{j}=\left\lfloor \beta *k_{j-1}\right\rfloor$                  **else** $V=\Phi *r_{i}$,                        $M_{a}=\max_{1<n<N}\left|v_{n}\right|$, $M_{e}=\underset{1<n<N}{median}\left|v_{n}\right|$,                        $\gamma =\tau \left(a_{3}M_{a}+b_{3}M_{e}\right)/c_{3}$,                        $L_{i}=\left\{n:\left|v_{n}\right|>\gamma \right\}$,                        $F_{i}=F_{i-1}\cup L_{i}$, $k_{j}=\left|L_{i}\right|$                  **end if**            **else**                  $k_{j}=\left|L_{i}\right|$            **end if**                  $j=j+1,K_{j}=K_{j-1}+k_{j}$      **else**            $F_{i}=F$      **end if****output**:$\overset{\wedge }{X}=\Phi_{F}^{+}y$


## 5. Simulation Results

The performance of SAVSMP algorithm is evaluated via a comparison with SAMP, SASP, OMP, and AStMP algorithms. A Gaussian sparse signal is used, the length N = 256, the sparsity P = 44 and the number of measurements M = 128. The sensing matrix $\Phi_{M\times N}$ is a Gaussian random matrix with σ=1, μ = 0. The parameters of $\sigma_{1}$, $\sigma_{2}$, τ take the value of $1\times 10^{-6}$, 10, 0.9, respectively. The background noise is Gaussian white noise, and the array is a uniform linear array composed of nine elements, and the array element spacing is half of the wavelength. All simulations are implemented in Matlab 2011a on the PC with 3.50 GHz Intel Core i3 processor and 4.0 GB memory in the Windows 7 system.

### 5.1. Sparsity Estimation

The initial sparsity estimation $K_{0}^{{}^{\prime }}$ of the SAVSMP algorithm is shown in Figure 5. Taking a smaller value for $\delta_{P}$ will lead to overestimation, whereas taking a larger value for $\delta_{P}$ will lead to underestimation. The SAVSMP algorithm adjusts $K_{0}^{{}^{\prime }}$ to $K_{0}$, and $K_{0}=(a_{1}*\delta_{P}+b_{1})*K_{{}_{0}}^{{}^{\prime }}$. As a result, the over-estimated value is reduced and the under-estimated value is increased. For example, $K_{0}^{{}^{\prime }}$ is halved when $\delta_{P}$ takes the value of 0.1 and tripled when $\delta_{P}$ takes the value of 0.35. However, $K_{0}^{{}^{\prime }}$ is halved whatever value the RIP constant $\delta_{P}$ takes in AStMP algorithm.

Estimated sparsity $K_{0}$ after treatment of the SAVSMP and AStMP algorithms are shown in Figure 6. Compared with the AStMP algorithm, the SAVSMP algorithm overcomes the under-estimation and over-estimation of initial sparsity estimation which can easily lead to higher iteration times and lower reconstruction accuracy. In addition, the SAVSMP algorithm eliminates the large influence of the RIP parameter $\delta_{P}$ on the estimated value, which mainly focuses on values between 20 and 25 whatever $\delta_{P}$ is.

### 5.2. Iteration Times

From the analysis of the proposed algorithm, the smaller the iteration times, the lower the computational complexity can be deduced. The iteration times comparison of different algorithms are shown in Figure 7. The SAVSMP algorithm needs 6 and 8 iterations, respectively, while $\delta_{P}$ takes the smaller value of 0.1 and larger value of 0.35. However, taking the value of 0.1 or 0.35 for $\delta_{P}$ has implications in the AStMP algorithm, which needs 8 or 12 iterations, respectively. The SASP algorithm needs 18 iterations, which is nearly three times that of the SAVSMP algorithm. The SAMP algorithm requires 23 iterations, which is the greatest number compared with these algorithms. Therefore, the iteration times of the SAVSMP algorithm is minimal, and $\delta_{P}$ has little influence on the iteration times.

### 5.3. Reconstruction Accuracy

To ensure the validity of the experimental data, 500 independent simulations are employed to calculate the reconstruction probability of the OMP, ROMP, StOMP, SP, CoSaMP, SAMP, SASP, AStMP, and SAVSMP algorithms.

The probability of exact reconstruction versus the sparsity level P of different algorithms is shown in Figure 8. It is shown that the reconstruction probability decrease as P increases. Compared with other algorithms, the OMP algorithm performs the worst. Although ROMP algorithm is more accurate than OMP algorithm, it is much lower than the StOMP, SP, and CoSaMP algorithms, which can achieve 100% reconstruction accuracy when P is small and drop dramatically when P⩾35. The reconstruction probability of the SAMP, SASP, AStMP, and SAVSMP algorithms is close when P is small; however, the SAVSMP algorithm performs better than other algorithms when P is greater than 60.

The reconstruction probability versus the number of measurements M of different algorithms is shown in Figure 9. It can be seen that the reconstruction probability increase as M increases. Compared with other algorithms, the ROMP algorithm performs the worst. Although OMP algorithm is more accurate than ROMP algorithm, it is much lower than the StOMP, SP and CoSaMP algorithms, which can achieve 100% reconstruction accuracy when M is larger than 95. The precision of the SAMP, SASP, AStMP, and SAVSMP algorithms are close when M is larger than 80; however, there construction probability of the SAVSMP algorithm is higher than other algorithms when P is smaller than 75. Therefore, SAVSMP algorithm shows obvious advantages compared with other algorithms in terms of reconstruction probability.

The mean square error (MSE) versus the signal-to-noise ratio (SNR) of different algorithms, such as the SAMP, SASP, AStMP, and SAVSMP algorithm, is shown in Figure 10. It is shown that the reconstruction error of each algorithm decreases gradually with the increase of SNR. The MSE of SAVSMP algorithm is less than other algorithms, and the performance advantage is more obvious with the increase of signal-to-noise ratio. It can be seen that the proposed algorithm has certain robustness.

### 5.4. Underwater Acoustic Signal Reconstruction Experiment

To verify the applicability of the algorithm to non-Gaussian sparse signals, this section uses underwater acoustic signal radiated by the ship to test the theoretical application value of SAVSMP algorithm. The ship-radiated noise obeying non Gaussian distribution consists of line spectrum and continuous spectrum.

Suppose that the propeller blades number is 3, the speed is 420 r/min, and the shaft frequency is $f_{z}=420/60=7$ Hz. Since the basic frequency is set as 21 Hz, the second harmonic, the third harmonic, and the fourth harmonic are 42 Hz, 63 Hz, and 84 Hz, respectively [17]. In addition, on the premise of satisfying the acoustic characteristics of the line spectrum, 150 Hz, 450 Hz, and 800 Hz are selected from the range of 100–1000 Hz [30]. Amplitude values of the selected seven frequency points are set to 3.5, 3.1, 2.2, 1.7, 1.5, 2.2, and 2, respectively, and the initial phases in Radian are set to 0.1, 2, −1, 3, 1, 0, and −1, respectively. The continuous spectrum peak frequency is set as 900 Hz, then 11 discrete frequency points from 0–900 Hz and 5 discrete frequency points from 1000–5000 Hz are selected. The corresponding frequency response of the finite impulse response (FIR) filter can be obtained if the desired response is given. The continuous spectrum can be obtained by applying the white Gaussian noise to the filter. Adding the line spectrum and the continuous spectrum, the spectrum of the ship-radiated noise can be obtained.

The reconstruction of the ship-radiated noise by the SAMP, SASP, AStMP, and SAVSMP algorithms are shown in Figure 11. Obviously, the SAVSMP algorithm can achieve underwater acoustic signal reconstruction with higher precision.

In addition to the reconstructed signal contrast, the peak signal-to-noise ratio (PSNR) will be used to evaluate the quality of the reconstruction signal. The PSNR is expressed as [31]
(16)$$PSNR=10\log\left[\frac{255^{2}*Framesize}{\sum_{n=1}^{Framesize}{\left(O_{n}-R_{n}\right)}^{2}}\right]$$
where Framesize represents the total number of pixels of the target signal, and $O_{n}$ and $R_{n}$ represent the nth pixel value of the original signal and the reconstructed signal, respectively. From Equation (Equation 16), we can see that larger PSNR means higher reconstruction quality. The compression rations versus the PSNR of the four algorithms are shown in Figure 12. The figure reveals that the PSNR will increases when the compression ratio increases. Compared with other algorithms in the figure, the SAVSMP algorithm has the highest PSNR for the underwater acoustic signal reconstruction.

## 6. Conclusions

In this work, a novel sparsity adaptive algorithm SAVSMP for underwater acoustic signal reconstruction has been presented. Since the SAVSMP algorithm estimates the sparsity by RIP criterion, it overcomes the disadvantage of the previous algorithms which require the prior information on the signal sparsity. SAVSMP selects atoms into the support set with an adaptive step size that can converge to the real sparsity quickly. In order to avoid over-estimation or under-estimation, SAVSMP proceeds the initial sparsity estimation adaptively by curing fitting method. Both the theoretical analyse and the systematical simulations demonstrate that the algorithm can accurately reconstruct ordinary random signal with low complexity. Given the significant advantage of high PSNR, SAVSMP can be used in reconstruction for underwater acoustic signals received by the sonar system.

## Figures and Tables

**Figure 1 sensors-22-05018-f001:**
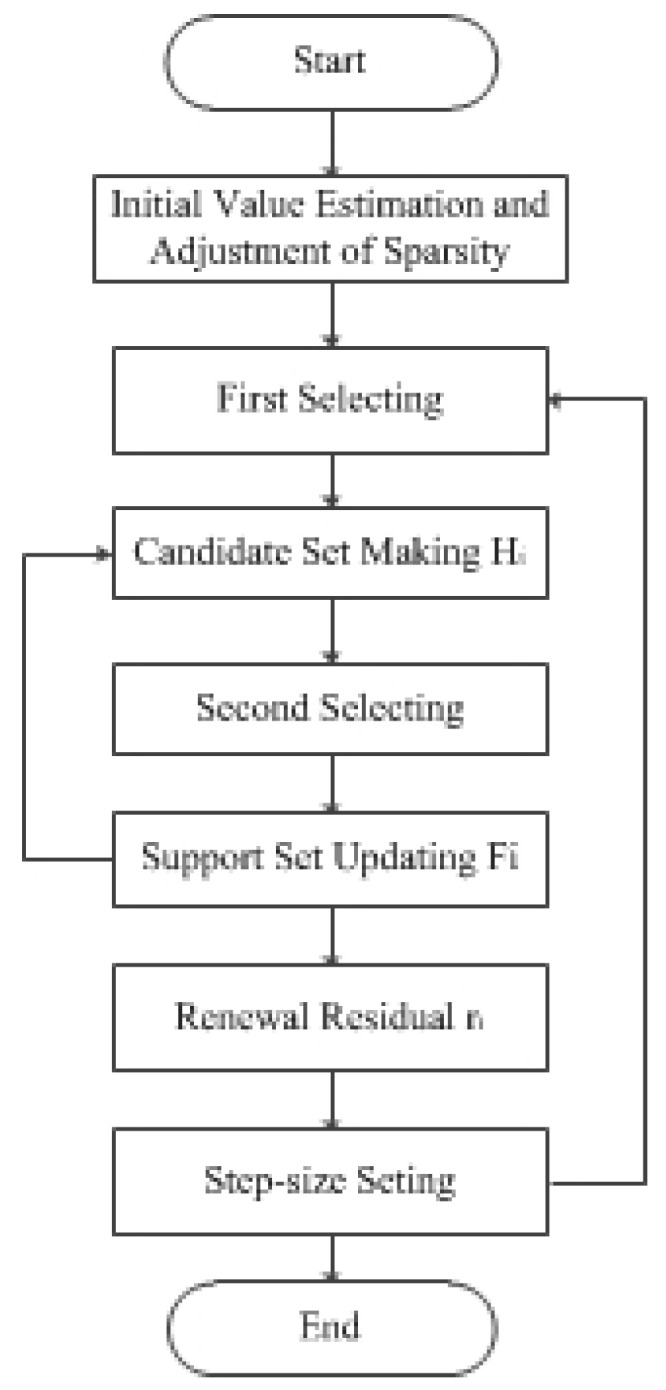
Flow chart of the main procedure.

**Figure 2 sensors-22-05018-f002:**
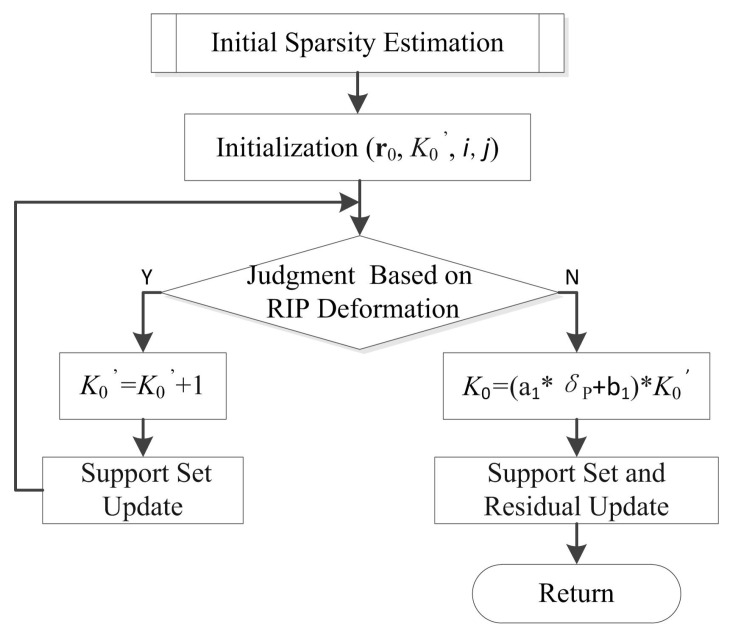
The flow chart of the initial sparsity estimation.

**Figure 3 sensors-22-05018-f003:**
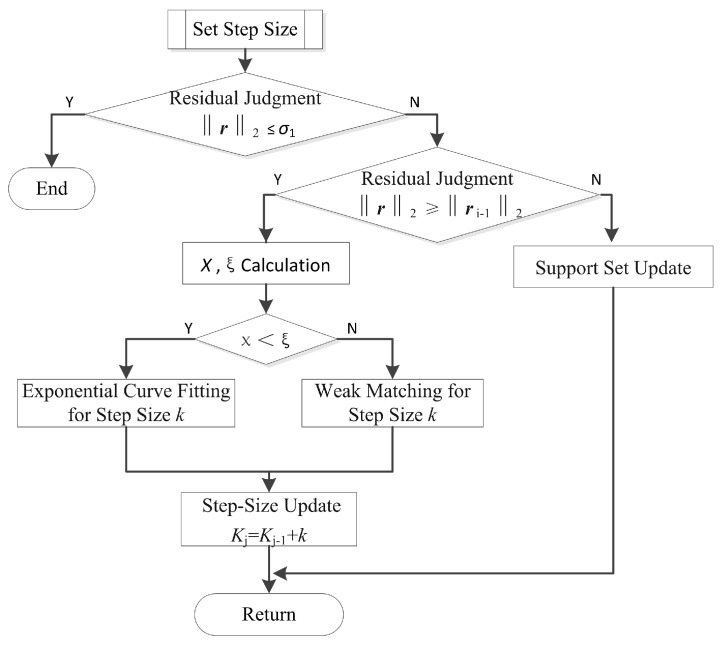
Flow chart of the step-size setting.

**Figure 4 sensors-22-05018-f004:**
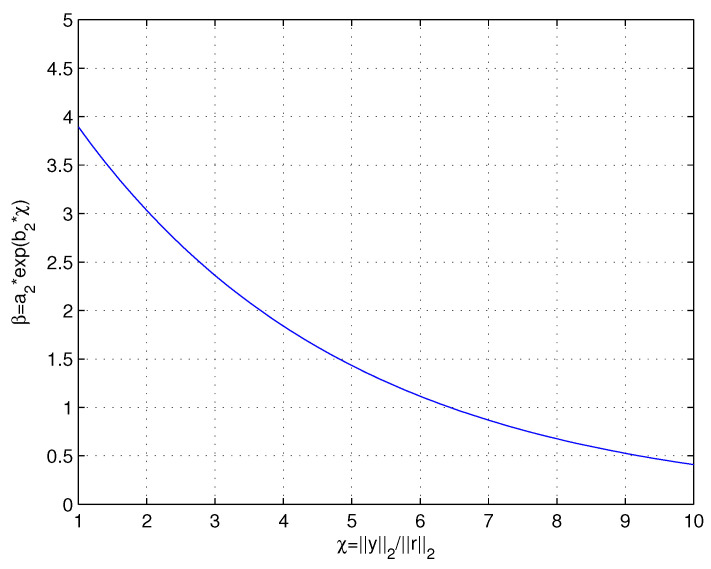
Measurementto residual energy ratio χ versus step-size scaling ratio β.

**Figure 5 sensors-22-05018-f005:**
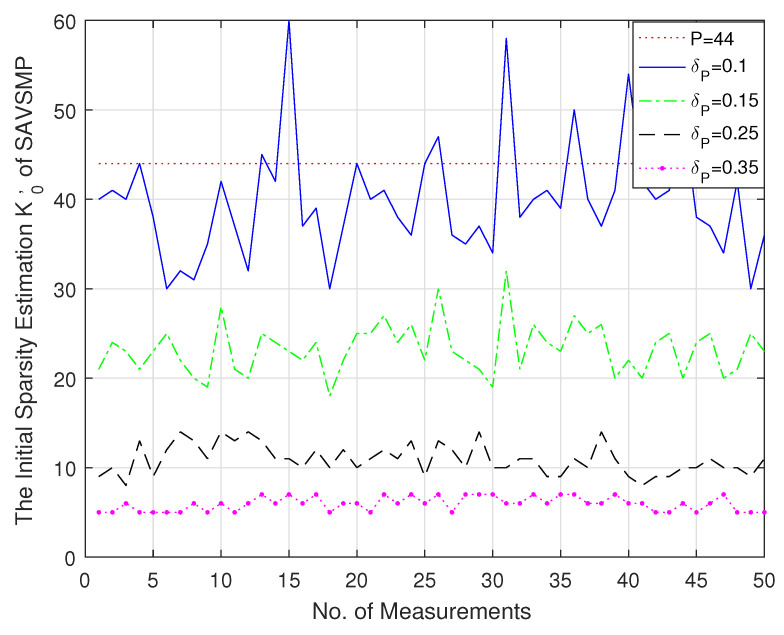
The number of measurements versus the initial sparsity estimation (M = 128, N = 256, P = 44 (the red dotted line in the figure)).

**Figure 6 sensors-22-05018-f006:**
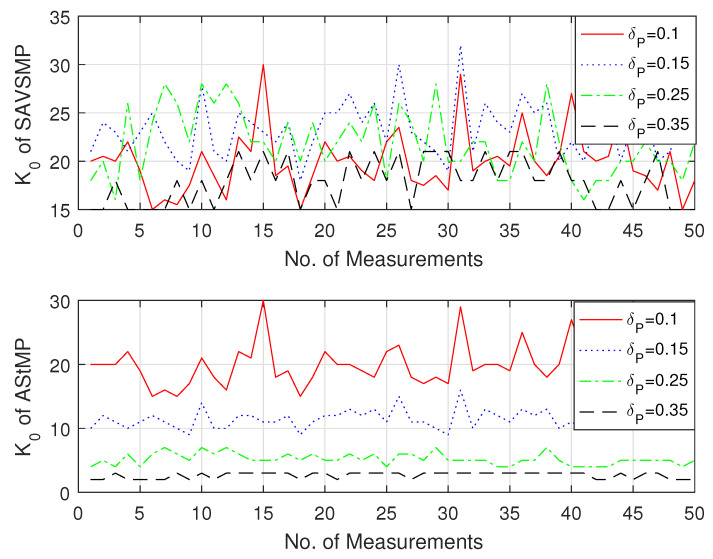
The estimated sparsity K0 after treatment of the SAVSMP and AStMP algorithms (M = 128, N = 256, P = 44).

**Figure 7 sensors-22-05018-f007:**
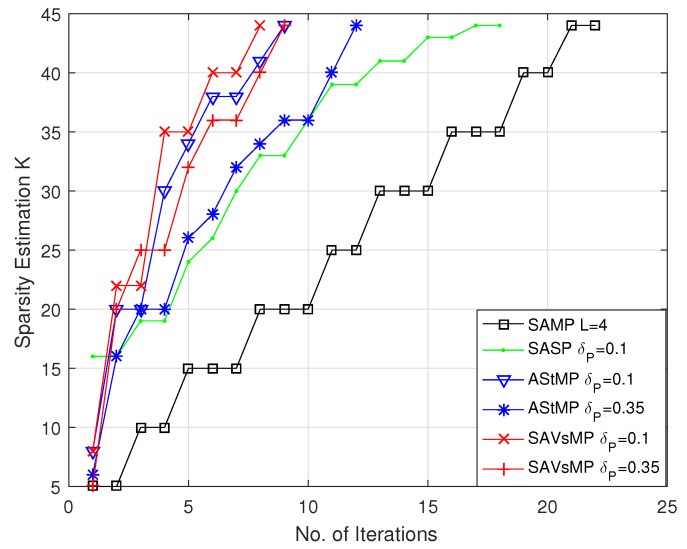
Iteration times comparison of different algorithms (M = 128, N = 256, P = 44).

**Figure 8 sensors-22-05018-f008:**
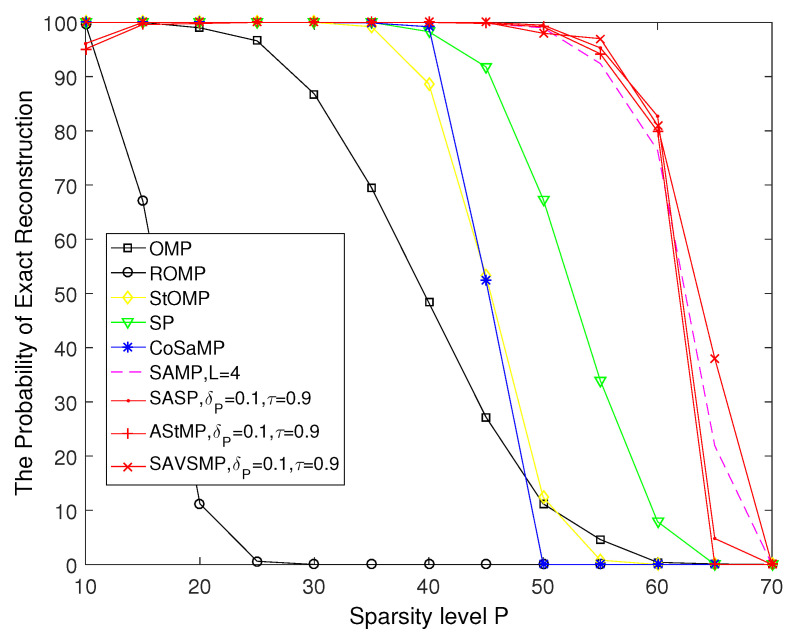
The probability of exact reconstruction versus the sparsity level P of different algorithms (M = 128, N = 256).

**Figure 9 sensors-22-05018-f009:**
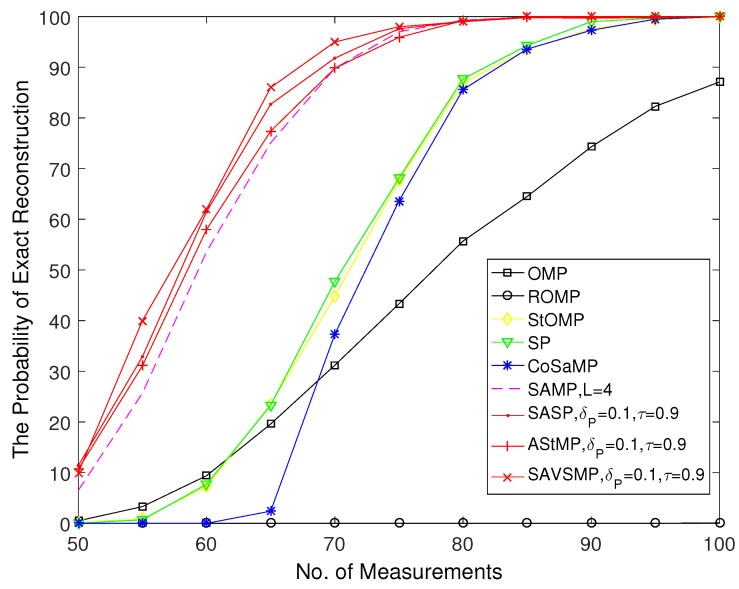
The reconstruction probability versus the number of measurements (M = 128, N = 256, P = 44).

**Figure 10 sensors-22-05018-f010:**
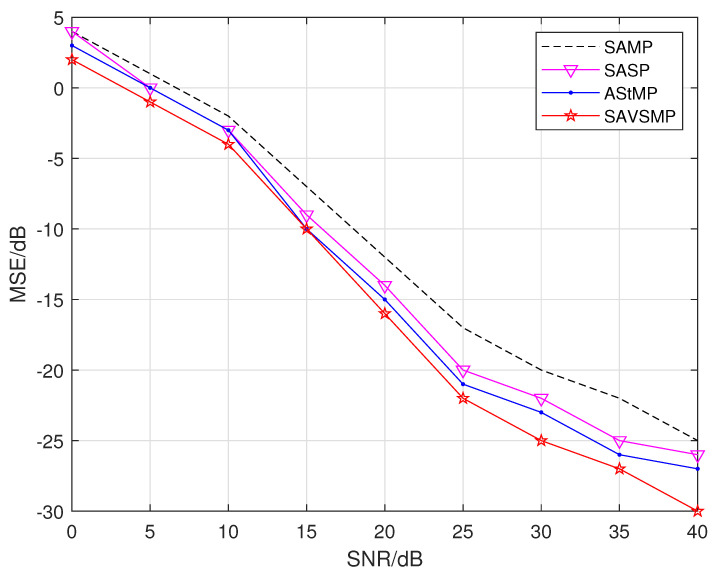
The reconstruction error to SNR of different algorithms (M = 128, N = 256, P = 44, $\sigma =1\times 10^{-6}$, ϵ=0.3, β=0.1).

**Figure 11 sensors-22-05018-f011:**
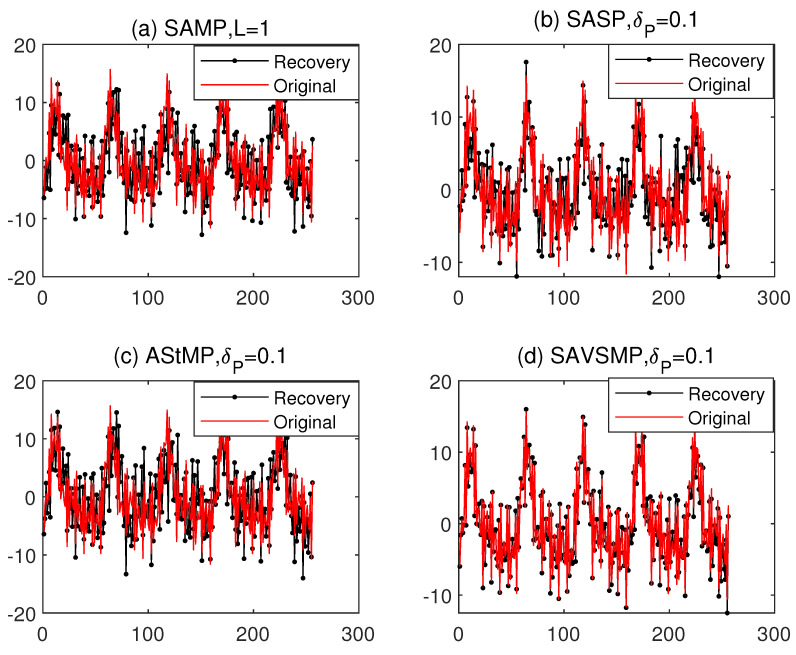
The reconstruction of ship-radiated noise.

**Figure 12 sensors-22-05018-f012:**
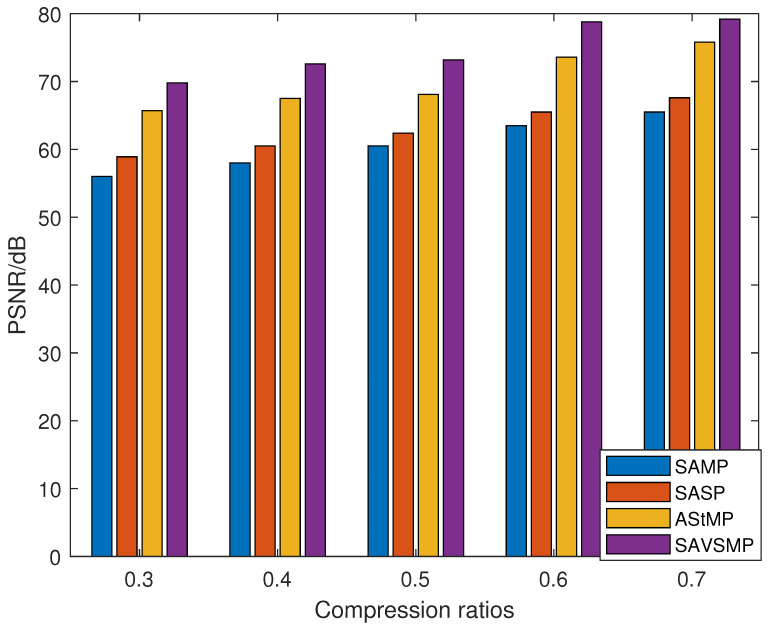
The compression rations versus the PSNR (underwater acoustic signal).

## Data Availability

Not applicable.

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
