# Peer review of "A Novel Sparsity Adaptive Algorithm for Underwater Acoustic Signal Reconstruction"

_sensors, 2022, doi:10.3390/s22135018_

Round 1
Reviewer 1 Report
This manuscript provides an interesting approach for reconstructing an underwater acoustic signal from compressed data based on the known sparsity. The RIP criterion is first used to estimate the starting value of sparsity and then curve fitting is applied to adjust the initial value. The experimental results are promising and are able to overcome the limitations of existing methods. I think the paper can be published once the authors consider my concerns, mainly regarding adding more details and explanations.
1. Figure 1 is not clear and should be replaced with a high-resolution image.
2. The authors have mentioned compressed sensing as one of the methods for reconstruction. Have the authors come across methods like deep learning that can be used for underwater signal reconstruction?
3. I did not find any analysis on how robust is the SAVSM algorithm to the measurement noise?
4. How do the sensor locations determine the accuracy of the proposed algorithm? How are the authors selecting sensor locations for their experiments?
Minor comments:
1. Line 10: high computation time
2. Line 59: In this paper,
3. Line 149: with the idea of
Reviewer 2 Report
In this paper, a novel sparsity adaptive algorithm SAVSMP for underwater acoustic signal reconstruction has been presented. In general, the work in this paper is well described. However, some discussions should be conducted before acceptance.
1. On line 98: the reviewer wanders to know how to determine a1 and b1. The authors should clearly clarify this in detail.
2. In simulations, The Gaussian sparse signal is used. The reviewer wanders to know whether the authors’ method is just suitable for Gaussian sparse signal. The authors should conduct more experiments by using different signals.
3. The authors should improve their English.
Reviewer 3 Report
There are a lot of written mistakes. They should be omitted. The presentation need to be improved - it is mentioned "this dissertation", a lot of "wrong" words, etc.
The research is actual, but some paragraphs are to be deleted - for example the presented software "code" is full of artificial symbols, it is not a contribution, etc.
The clear dividing and statement of the originality will strongly improve the quality of the paper.
Round 2
Reviewer 2 Report
The paper is improved after revision, and it is suggested to be accepted.